# Contour-Based Wild Animal Instance Segmentation Using a Few-Shot Detector

**DOI:** 10.3390/ani12151980

**Published:** 2022-08-04

**Authors:** Jiaxi Tang, Yaqin Zhao, Liqi Feng, Wenxuan Zhao

**Affiliations:** College of Mechanical and Electronic Engineering, Nanjing Forestry University, Nanjing 210037, China

**Keywords:** wild animal instance segmentation, few-shot object detection, contour approximation, deep learning

## Abstract

**Simple Summary:**

Biodiversity monitoring is one of the primary means of ecological research. With the development of convolutional neural networks (CNNs) in the field of instance segmentation, CNNs are also used for species recognition. Almost all species recognition models apply pixel-based instance segmentation to recognize animal species. However, pixel-based instance segmentation models require a large number of annotations and labels, which makes them time-consuming and unsuitable for small datasets. Therefore, in this paper, we propose a contour-based wild animal instance segmentation model that can reach a balance between accuracy and real-time performance.

**Abstract:**

Camera traps are widely used in wildlife research, conservation, and management, and abundant images are acquired every day. Efficient real-time instance segmentation networks can help ecologists label and study wild animals. However, existing deep convolutional neural networks require a large number of annotations and labels, which makes them unsuitable for small datasets. In this paper, we propose a two-stage method for the instance segmentation of wildlife, including object detection and contour approximation. In the object detection stage, we use FSOD (few-shot object detection) to recognize animal species and detect the initial bounding boxes of animals. In the case of a small wildlife dataset, this method may improve the generalization ability of the wild animal species recognition and even identify new species that only have a small number of training samples. In the second stage, deep snake is used as the contour approximation model for the instance segmentation of wild mammals. The initial bounding boxes generated in the first stage are input to deep snake to approximate the contours of the animal bodies. The model fuses the advantages of detecting new species and real-time instance segmentation. The experimental results show that the proposed method is more suitable for wild animal instance segmentation, in comparison with pixel-wise segmentation methods. In particular, the proposed method shows a better performance when facing challenging images.

## 1. Introduction

The diversity monitoring of wildlife is important in the research, conservation, and management of wildlife. Most natural reserves are equipped with camera traps to monitor wildlife species and their behaviors. The images obtained include a wealth of significant information regarding wildlife, such as species composition, individual behaviors, and population dynamics. However, these data mainly depend on manual screening, which cannot keep up with the speed of image accumulation. As a result, the use of data in research, conservation, and management is seriously restricted.

Masking animal objects from a large number of wildlife images can not only make it possible to study the species composition of an ecosystem but is also the foundation of wildlife re-identification and behavior tracking. With the help of supervised machine learning, early work [1,2] has reduced time and labor costs to some extent and improved the efficiency of screening and identification, but these methods still have two flaws. First, they need to be specially customized for specific problems. Worse still, the customization might result in an exclusive model for a certain dataset. Second, the limited amount of training data and lack of computing power weaken the generalization capacity of these models.

There exist many deep-learning-based object detection methods, and they have been applied in many fields, such as autonomous driving [3], facial recognition [4], defect detection [5,6], and medical imaging [7,8]; the historical champion network ResNet from the ImageNet LSVRC (Large-Scale Visual Recognition Challenge) competition outperformed the human recognition level [9]. Models based on CNNs have shown good accuracy and robustness for recognition of certain species. These models are used to help ecologists with labeling and detecting animals. Some researchers directly apply object detection networks, such as ResNet18 [10], GoogLeNet [11], Faster R-CNN [12,13,14], YOLO series [15,16,17], and AlexNet [18], for species recognition. To achieve better performance, a few researchers have tried to improve the networks or fused several networks. For example, Emmanuel et al. (2016) [19] shrank the number of neurons in both fully connected layers and the last inception layer, thus modifying the AlexNet and GoogLeNet architectures. Sara Beery et al. (2020) [20] incorporated an attention mechanism into the context R-CNN for detecting species in images captured by camera traps.

Traditional image segmentation methods, such as thresholding [21] and edge detection [22], can only detect objects whose visual features have great differences from their backgrounds. The principle of the thresholding method is to divide pixels into several regions by the threshold. However, its application scenarios are limited, thus making it suitable only for images with obvious gray differences. Edge-based segmentation methods are used to find the pixels at the boundary, and then connect them to segment the targets. These methods are simple and only require a little calculation, but when they face wild animal images with unclear grey features, complex backgrounds, and camouflaged environments, they cannot be used for the segmentation task. To avoid the above problems, researchers have applied CNNs in the field of instance segmentation. Among the previous research, the more classic Mask R-CNN [23] predicted the binary mask to achieve pixel-level segmentation. PANet [24] improved the feature hierarchies of networks and information flows between frameworks. YOLCAT [25] used ResNet and FPN as the backbone; additionally, they added a mask coefficient branch for generating instance masks. Shu Liu et al. (2017) [26] proposed a sequential grouping network (SGN). In [26], semantic segmentation was used to identify foreground pixels for instance segmentation. PolarMask [27] transformed the instance segmentation problem into the instance center classification. Hao Chen et al. (2020) [28] presented a one-stage dense instance segmentation method based on the detection model FCOS [29], which reduced the amount of computation. 

Since much effort has been made, with respect to CNNs, the networks have been constantly applied in species detection tasks. In fact, only a few valid and useful wild animal pictures are taken by camera traps because the cameras are triggered irregularly. However, the above-mentioned deep convolutional neural networks need a large number of annotations and labels, thus making them unsuitable for small datasets.

Transfer learning can be used to solve the problems of small datasets and has been proven to be an effective strategy. However, transfer learning first identifies the training data, then generalizes it to the test data. Therefore, transfer learning methods can recognize small sample objects that are similar to the objects in training sets, and their categories are included in the training sets, as shown in Figure 1a.

Unlike transfer learning methods, the few-shot object detection method (FSOD) based on CNNs [30] was first proposed to recognize novel objects by only training a small number of labeled samples in the training set. Some strategies were also presented to improve the performance of FSOD [31,32,33]. Few-shot learning involves the application of meta-learning in the field of supervised learning. As shown in Figure 1b, few-shot learning methods train the models (update the model parameters) on multiple subtasks, in order to make the models ‘learn to learn’. Furthermore, it can be seen from Figure 1 that the content of the testing and training tasks can be completely different. Therefore, few-shot learning methods can recognize new categories that are not included in the training sets of base classes. Conversely, transfer learning methods often recognize the categories that are included in training sets. After learning a large amount of data from a certain category, few-shot learning can quickly learn new categories with only a small number of samples. Because wild animal images are difficult to capture, especially for rare animals, we used the few-shot object detection method [34] to recognize a novel species without a large number of image samples for that species.

In this paper, we propose a two-stage method, for instance, the segmentation of wildlife, including object detection and contour approximation. In the object detection stage, FSOD [34] is not only used as the detector to generate the initial bounding boxes of animals, but it is also used to identify wild animal species. In the case of a small wildlife dataset, this method may improve the generalization ability, with respect to wild animal species recognition, and even identify new species that only have a small number of samples in the training set. In the second stage, the initial wildlife bounding box is regarded as an initial rectangle contour and is input to deep snake [35] to approximate the final contour of the animal shape. The contributions of this paper are threefold:We explore a novel two-stage model for instance segmentation of wild animals, which uses FSOD for animal species recognition and as the initial contour detector, and deep snake as a contour approximation model for the instance segmentation of wild mammals. The model combines the advantages of detecting new species and real-time instance segmentation.We fine-tune the FSOD convolutional neural network to recognize both the wildlife species in small datasets and new species that only have a small number of samples in the training set, which can solve the problems of imbalanced datasets and small datasets caused by the drawbacks of camera traps.We propose a contour-based wildlife instance segmentation strategy by selecting the optimal detector for the deep snake submodule, which can correct the error between the initial bounding box and actual localization of the wild animal by exploiting the cycle-graph structure of a contour. Due to the unnecessary classification of each pixel, the method is more suitable for real-time segmentation of wild animals.

## 2. Materials and Methods

In this section, we first introduce the customized dataset made for wild mammal research. This dataset can also be used to study mammalian species composition and for analysis. Then, we provide the pipeline of the proposed animal instance segmentation network. After that, we describe the species identification detector and animal contour approximation method.

### 2.1. Materials

#### 2.1.1. Configurations

All of our experiments were performed on a personal computer with a 1060Ti GPU and the Ubuntu 20.04 operating system. Additionally, CUDA v10.1 was applied to train the model, which accelerated the processing speed, and Python version 3.8.12 was used. Other required libraries we used were numpy (v1.21.2), Pytorch (v1.6.0), pycocotools, cython, matplotlib, and opencv.

For the detector submodule, we divided the 12 categories in the dataset into base classes and novel classes at a ratio of 3:1, in order to verify the generalization ability of this method without being unfair. Considering the computing power of our personal computer, we set the batch size to 2 and trained the model using SGD. The following parameters were used to train the submodule of the initial bounding box of an animal image, based on the few-shot method [34]. Thus, we adopted the same parameter settings as [34]. The momentum was 0.9, and the weight decay was 0.0001. A smaller learning rate should be set for small datasets. Due to the different numbers of samples of animal species in the dataset, the learning rate of the base classes and fine-tuning novel classes were different; these were 0.02 and 0.001, respectively.

For the contour approximation submodule, we sampled 40 points on the diamond contour for more context information. During the octagon contour deformation stage, N points were uniformly sampled on the octagon contour; correspondingly, N offsets were output. We took N as 128. It should be noted that, for points far away from the ground truth or for large offsets, regression is challenging, and we solved this problem by applying multiple iterations. The number of iterations was set to 3.

#### 2.1.2. Dataset

Existing public datasets commonly used for object detection include a limited number of animal images, especially with respect to wild animal species. For example, the PASCAL VOC dataset only contains 3 kinds of livestock (cows, horses, and sheep), and the COCO dataset includes limited mammals (elephants, bears, zebras, giraffes, etc.). 

In order to facilitate analysis and research, in relation to more wild mammal species, we collected images of six species of wild mammals from the public datasets COCO and PASCAL VOC (https://cocodataset.org, http://host.robots.ox.ac.uk/pascal/VOC/, accessed on 4 July 2022), and the images of another six species were obtained from wildlife documentaries. Finally, a balanced dataset, named MammData, was constructed. The dataset MammData includes 12 common wild mammal species, with a total of 4884 images, around 400 per species. All the images in the dataset are two-dimensional RGB color images with a single resolution of 1280 × 720. Some examples from the dataset are shown in Figure 2. This specific dataset contains various postures of mammals, as well as close-up animal pictures, for better robustness of species identification. We allocated the training and evaluation set at a ratio of 0.85, and the details of our dataset are shown in Table 1.

### 2.2. Network Structure

The pipeline of the wild mammal instance segmentation network is shown in Figure 3. The network is mainly composed of a detector submodule and contour approximation submodule. FSOD is adopted as the detector submodule, which is used for species identification and generation of the initial rectangular box used for contour approximation. Then, the rectangular box is processed as the input of the first deep snake block, which generates the octagon contour. Then, through multiple iterations of the second deep snake block, the box is gradually approximated, until the contour tightly outlines the animal shape.

#### 2.2.1. Detector Submodule

We applied the FSOD method [34] as a detector submodule. As shown in Figure 3, the detector submodule consists of ResNet101 as the backbone, region proposal network (RPN), and fully connected (FC) network after the ROI pooling. The box predictor is made up of a box classifier and box regressor, which are used to predict category and bounding box localization, respectively.

As shown in Figure 4, the training strategy is divided into two stages, called base training and fine-tuning. In the base training stage, FSOD randomly selects C categories and K samples of each category in the training set. A total of C∗K animal images comprise the base classes; that is, the base classes contain abundant animal images that are used to train the parameters of the model. However, in the fine-tuning stage, we only pick a small number of images from both base classes and novel classes, where no samples are included in the training set of base classes. At this stage, if each class in the training set of small datasets contains n animal images; it is named n-shot. The few-shot concept is embodied from here. 

It is thought that the backbone and RPN are provided for feature extraction, and they are, in other words, irrelevant to the classification. Therefore, the feature maps, parameters, and weights learned by these parts can be transplanted directly into novel classes without fine-tuning. Explicitly, all we need to do at the second stage is fine-tune the weights of the box predictor. We feed the network with abundant images in base classes at the first stage of training, in order to obtain features and weights that can then be used for novel classes. Afterwards, we create a balanced small subset, including both base and novel classes, for the fine-tuning stage. We start by assigning randomly initialized weights to the box predictor for novel classes and fine-tune it with a fixed feature extractor. That is, we train the ability of feature extraction using abundant animal images and use the learning ability to extract features of animal images in small sample classes or new classes (called few-shot). After that, we fine-tune the parameters of classification layers in the proposed model to realize few-shot animal image recognition. Therefore, this mechanism enables the model to learn the common parts of base classes, such as how to extract important features and compare the similarity of samples; thus, it can also be generalized to small sample classes.

#### 2.2.2. Contour Approximation Submodule

The detector submodule only obtains the rectangular box of wild animals; inevitably, there is a discrepancy between the bounding box and actual localization of the wild animal. To solve this problem, we introduced the contour approximation submodule deep snake [35], in order to gradually deform the rectangular box into an animal shape.

The deep Snake block [35,36,37,38] is a learning-based snake algorithm that was inspired by traditional snake algorithms, and it is a contour-based segmentation method. The deep snake algorithm regards the contour as a set of variables and optimizes these variables. It can approximate the contour to the object boundary by setting appropriate parameters. Considering that the contour is a cycle graph, deep snake applies circular convolution to learn the features on the contour and realize approximation.

The detailed contour approximation submodule is shown in Figure 5. We set both the box information and corresponding image as input. First of all, we transform the rectangular box into a diamond contour by connecting the midpoints of the four lines of the box. The deep snake block [35] takes this diamond contour as input and then predicts four extreme points surrounding to the top, leftmost, bottom, and rightmost of the animal body. Inspired by [36,37,38], extreme points on the object can provide more contextual information than bounding boxes and improve the efficiency of segmentation. We create the octagon contour based on these four extreme points. We extend 1/4 of the length of the corresponding edge, without exceeding the box boundary. The eight endpoints created are then regarded as eight vertices of the octagon contour. The second deep snake block processes this octagon by evenly sampling N points on it and outputs a much closer contour to the animal body. The octagon will gradually approach the shape of the animal body after multiple iterations. Using this kind of multiple deforming box (or contour) method to approximate the shape of an animal body can also correct the error that occurs at the former detector submodule to some extent.

## 3. Results

In this section, we describe using our dataset, MammData, to evaluate the performance of the proposed network. First, we conducted fine-tuning experiments on the training sets of the different novel classes containing different numbers of animal images and selected the optimal number of animal images for the training. Then, we tested the species recognition ability for some challenging images, including incomplete bodies, overlapped multiple animals, small animals, and camouflaged animals. Finally, we compared the performance of the proposed detector submodule with that of the state-of-the-art object classification networks.

In all experiments, we used AP (average precision) (https://cocodataset.org/#detection-eval, accessed on 4 July 2022) as the evaluation metric of the model. AP means the average precision over all categories. Generally, the higher the AP score, the better the classification model. AP50 and AP75 mean that the matching threshold IoU (intersection over union) values are 0.5 and 0.75, respectively. The higher the threshold is set, the greater the challenge to the model. IoU is defined as the area of overlap between the detected animal region and ground truth divided by the area of union between the detected animal region and ground truth.
(1)$$IoU=\frac{groundTruth\cap prediction}{groundTruth\cup prediction}$$

### 3.1. Detecting New Species

In the fine-tuning stage, if the training set of each class contains n animal images, it is named n-shot. For instance, 5-shot means 5 images of each class in the sample classes. Therefore, we evaluated the performance of detecting the animal species in small sample training sets. We conducted fine-tuning experiments for 1-shot, 2-shot, 3-shot, 5-shot, 10-shot, and 30-shot. The total loss for n-shot fine-tuning is shown in Figure 6. As shown in Figure 4, the fine-tuning can converge quickly during the training iterations. The performance of fine-tuning is a little worse with the decrease in the number of samples in the base training set, which proves the feasibility of this method.

The comparison results of average precision (AP), AP50, and AP75 between novel classes of fine-tuning and base training are shown in Figure 7. As can be seen in the figure, the proposed method performs well when training on both the base and novel classes, which shows that the features trained on the base classes are effectively transferred to the novel classes. 

We also tested the fine-tuning model for all shots, and the test results are shown in Figure 8. In general, the 10- and 30-shot models have higher accuracy, and the 30-shot model has the best prediction results for novel classes, including category prediction and box regression. The 1- and 2-shot models can only correctly detect a small number of novel classes, due to too few training samples, while the 3- and 5-shot models can detect most of the novel classes; however, the accuracy of classification is low, and the accuracy of box regression is less than for the 10- and 30-shot models. Therefore, the proposed detector submodule can detect new species that are not included in the training set of the base classes.

### 3.2. Recognizing Multiple Species

We also tested the performance of the proposed method in recognizing multiple species based on the metrics AP, AP50, and AP75. Twelve wild mammal species in the dataset MammData were used for the comparison of the proposed method and other state-of-the-art methods. As shown in Table 2, the proposed method significantly outperforms YOLOv4 and FCOS. The AP value is slightly lower, and the metric values AP50 and AP75 of the proposed method are higher than Sparse R-CNN. The dataset MammData contains many challenging images, as shown in Figure 9. The challenging images include multiple small targets, occluded or overlapped animal bodies, camouflaged backgrounds, and incomplete bodies, which increases the difficulty of species identification. Traditional object detection models based on CNNs, such as YOLOv4, FCOS, and CenterNet, cannot accurately detect animal objects in challenging images; thus, as seen in Table 2, the three models have lower values of AP, AP50, and AP75. 

In particular, the proposed method and Sparse R-CNN are capable of species identification for challenging images; however, as shown in Figure 7, the Sparser R-CNN method cannot detect the small animals in an image containing multiple animals. Sparse R-CNN also has higher errors in marking the rectangle box than the proposed method when animal objects are in complex backgrounds (occluded, camouflaged, overlapped) or only a part of the animal body appears in an image.

### 3.3. Segmenting Animal Objects

To evaluate the performance of the contour approximation model, we adopted state-of-the-art instance segmentation models for our comparison. The comparison results for the animal instance segmentation networks are shown in Table 3. As shown in Table 3, our model and PANet achieved higher AP and AP50. This is because SGN [26] and Mask-RCNN [23] sometimes miss small objects in animal images and incorrectly segment one animal into two parts. In contrast to the pixel-based representation networks PANet [24] and SGN [26], our contour-based animal instance segmentation network is not limited within a bounding box and has fewer parameters. Therefore, our model is the fastest among these methods, and is almost five times faster than PANet.

In the deep snake network [35], CenterNet is used as the detector of initial rectangular boxes. Thus, we evaluated the performance of the proposed method with deep snake [35] and CenterNet with deep snake [35]. The comparison results are shown in Figure 8. For multiple animals or overlapped objects, the box provided by the proposed method is more accurate and complete; thus, the result of its contour approximation is almost close to the ground truth. However, the CenterNet box appears to mark two elephants together, and the box regression position is inaccurate. Because only half of the body is shown in this image, CenterNet cannot predict this box. After being processed by the deep snake block, these results are not ideal. As shown in Figure 10, CenterNet has regressed two very similar boxes for the same horse, and the results of contour approximation almost overlap. This will have an impact on the subsequent data processing, such as species quantity prediction, behavior recognition, etc. In addition, the FSOD method detected two cheetahs; because the cheetahs are in a camouflaged environment, CenterNet only returned one box for the two cheetahs, which led to errors in contour approximation. The results were similar for the cow. Many comparison results prove that the regression accuracy of the box directly affects the contour approximation accuracy of the deep snake block, while the box output by the proposed method has higher accuracy, so the contour approximation results are more ideal.

## 4. Conclusions

In this paper, we proposed a two-stage method for the instance segmentation of wildlife, including object detection and contour approximation. We replaced the detector CenterNet in the deep snake convolutional network with FSOD to detect wildlife species and segment animal instances more accurately. The proposed method can also detect rare species with few samples. The proposed model combines the advantages of detecting new species and real-time instance segmentation. We first constructed the mammal dataset MammData and labeled images for species recognition. The experimental results show that the regression accuracy of the box directly affects the contour approximation accuracy of the deep snake block, while the box output by the proposed method has a higher accuracy, meaning that the contour approximation results are more ideal.

Although we successfully combined the FSOD and deep snake methods and our method can detect rare animals from few examples, there are still some improvements that can be made. In the future, we will collect more images of different kinds of wild mammals to expand our dataset. We found that the box regression results for novel classes are good, but the classification accuracy for novel classes is not as good as the box regression results. Therefore, we will focus on the network structure of FSOD to solve this problem. In addition, this method can only detect static images; it is not applicable to videos. We will try to apply novel video object tracking models to the instance segmentation of wildlife animal videos.

## Figures and Tables

**Figure 1 animals-12-01980-f001:**
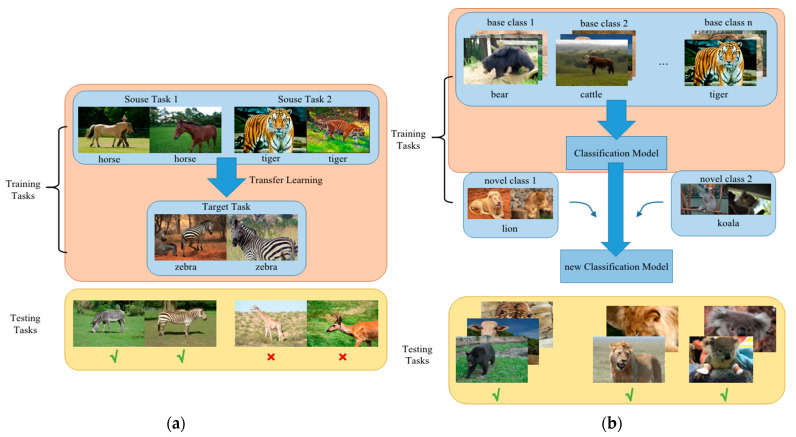
Transfer learning and few-shot learning. The transfer learning method is shown as (**a**), and the few-shot learning method is shown as (**b**). Transfer learning methods recognize small sample objects that are similar to the objects in training sets, and their categories are included in training sets. Few-shot learning methods train the models (update the model parameters) on multiple subtasks, in order to make the models ‘learn to learn’; the content of the testing and training tasks can be completely different.

**Figure 2 animals-12-01980-f002:**
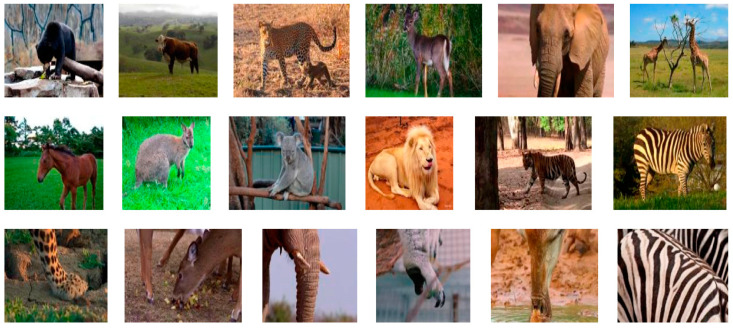
Examples of the dataset. The first two rows display one example of each mammal species in the dataset. The third row shows incomplete body parts, such as an elephant’s trunk, zebra stripes, cheetah leg, and so on.

**Figure 3 animals-12-01980-f003:**
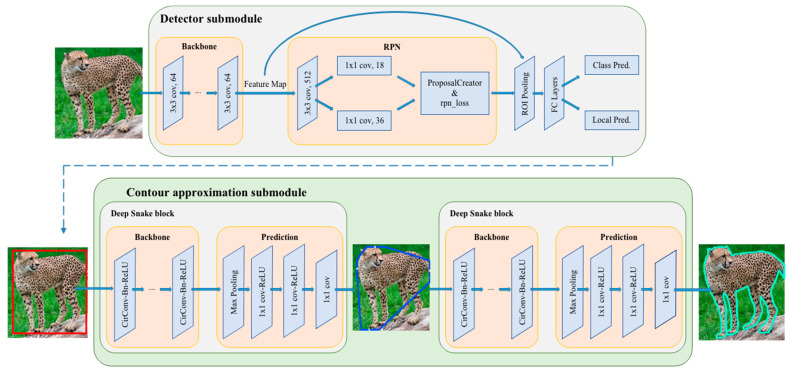
Overall illustration of the proposed two-stage wildlife instance segmentation method. The detector submodule contains the backbone ResNet101, followed by the region proposal network (RPN), and fully connected (FC) layers propose object categories and bounding box localization. The contour approximation submodule consists of two deep snake blocks. The first deep snake block turns the rectangular box generated by the detector submodule into the octagon contour, and the second deep snake block takes the octagon contour as the input to approximate the contour of the animal shape.

**Figure 4 animals-12-01980-f004:**
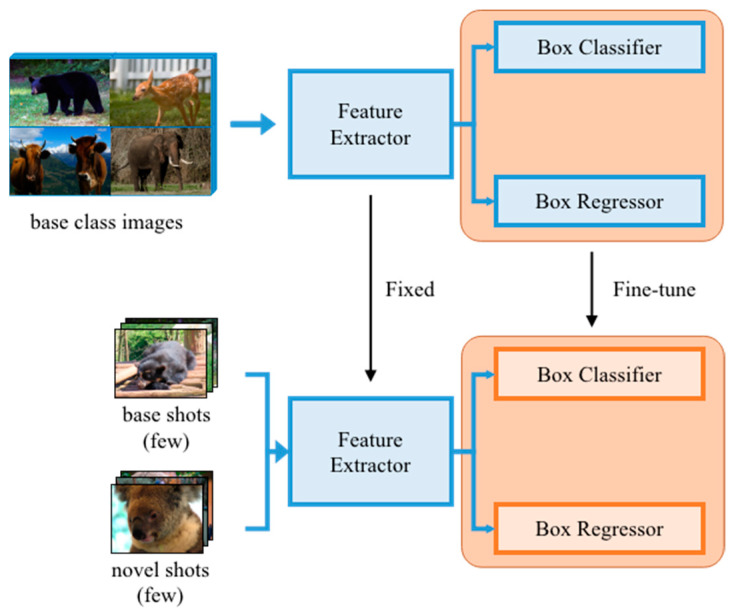
The proposed two-stage FSOD training strategy. In the base training stage, we apply abundant images to train the detector, including the feature extractor, box classifier, and box regressor. In the fine-tuning stage, the feature extractor is fixed, and we only select a few images to fine-tune the box classifier and box regressor.

**Figure 5 animals-12-01980-f005:**
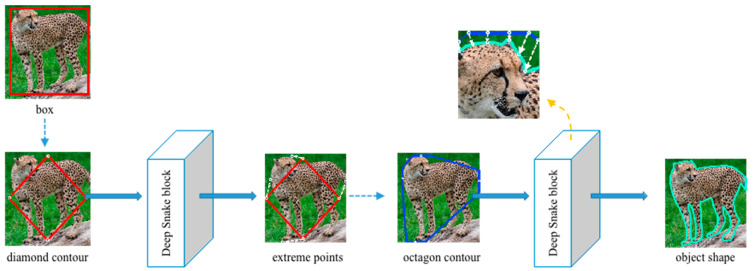
Detailed structure of contour approximation sub-module. We first import the diamond contour formed from the rectangular box into the deep snake block, which outputs the offsets to extreme points. Then, we adjust these extreme points to adapt the contour approximation and put them into the deep snake block again. After multiple iterations, the box finally surrounds the object shape tightly.

**Figure 6 animals-12-01980-f006:**
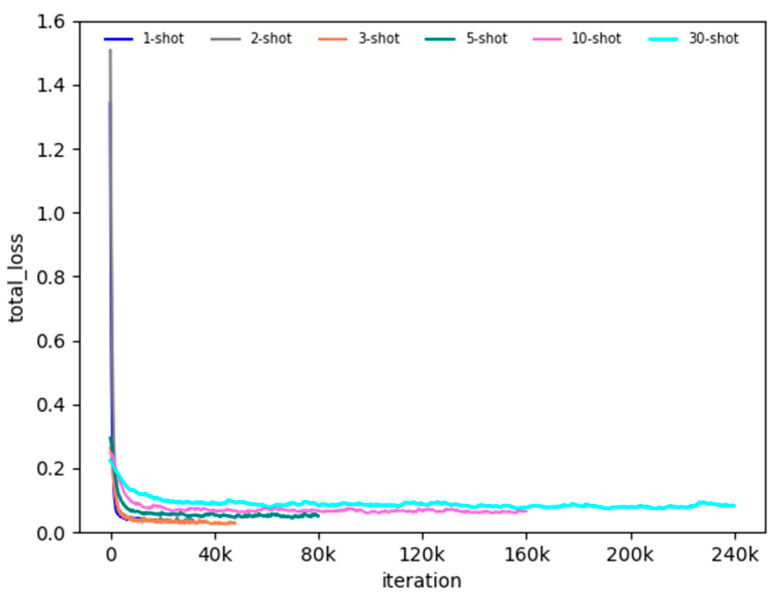
The total loss of n-shot fine-tuning.

**Figure 7 animals-12-01980-f007:**
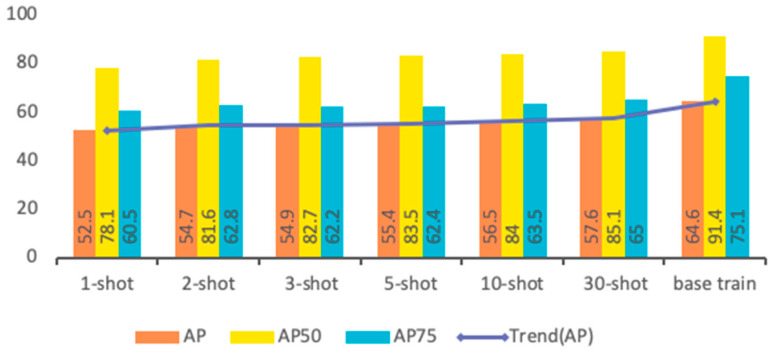
Comparison of AP, AP50, and AP75 between n-shot fine-tuning and base classes of training.

**Figure 8 animals-12-01980-f008:**
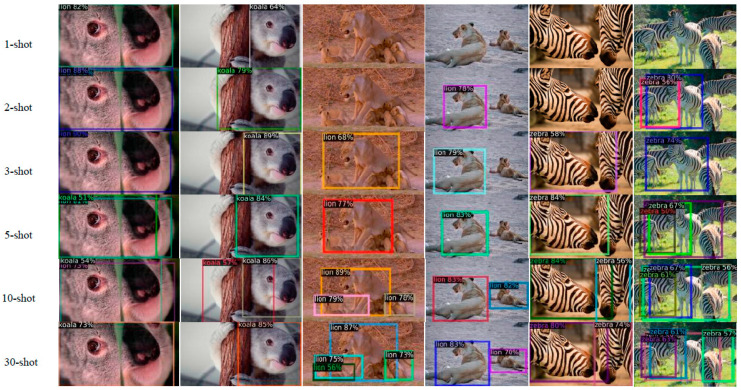
Results for the fine-tuning model of novel classes. In general, the 10- and 30-shot models perform best on both box regression and classification. The 1- and 2-shot models can only detect a few novel classes, because the training samples are too few to train a good model.

**Figure 9 animals-12-01980-f009:**
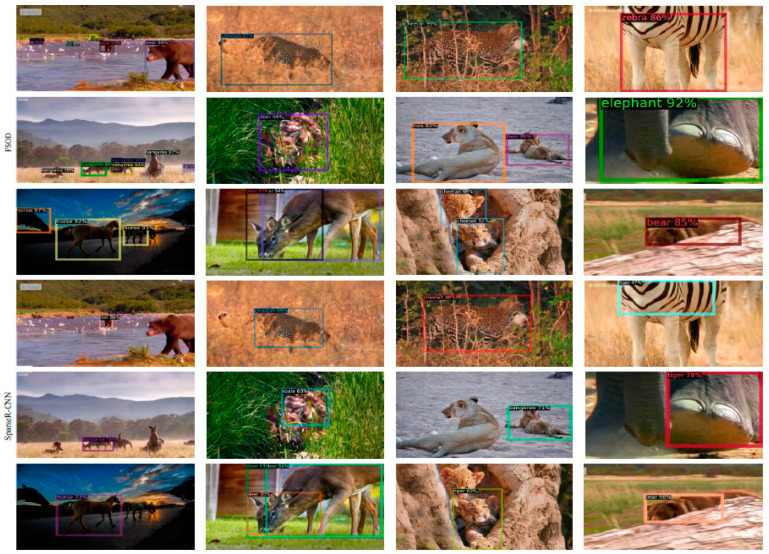
Challenging example image results for the proposed method and Sparse R-CNN method. First column: multiple animals; second column: occluded or overlapped; third column: camouflaged; fourth column: incomplete body.

**Figure 10 animals-12-01980-f010:**
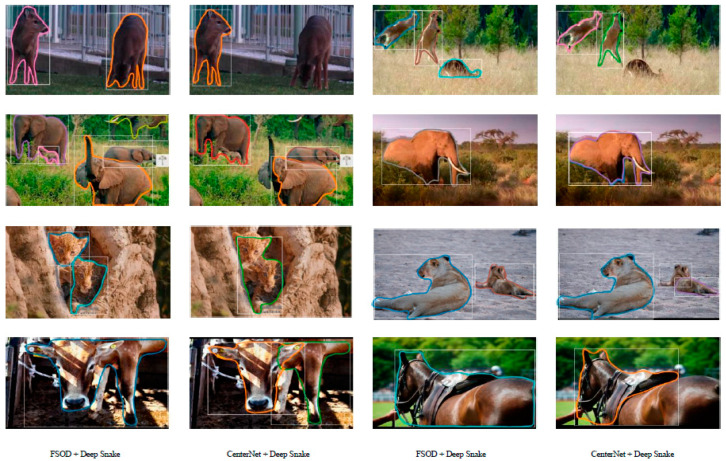
Contour approximation comparison between the proposed method and the CenterNet method. First row: multiple animals; second row: occluded or overlapped; third row: camouflaged; fourth row: incomplete body. For the above challenging images, the proposed model can detect animal contours correctly. However, CenterNet encountered some problems, such as missing the animals, regressing two animals into a contour, or dividing an animal into two parts.

**Table 1 animals-12-01980-t001:** Composition of the dataset MammData.

Class Name	Number of Images	Number of Training Instances	Number of Validation Instances
Bear	415	543	101
Cow	351	456	88
Cheetah	471	452	79
Deer	404	454	81
Elephant	404	595	100
Giraffe	438	520	79
Horse	400	482	82
Kangaroo	367	510	60
Koala	438	383	80
Lion	397	497	92
Tiger	399	413	72
Zebra	400	578	80
Total	4884	5883	994

**Table 2 animals-12-01980-t002:** Comparison of AP, AP50, and AP75 results for the MammData dataset.

Methods	Backbone	AP	AP50	AP75
Sparse R-CNN [39]	ResNet-50	58.7	78.3	64.6
YOLOv4 [40]	DarkNet-53	19.8	32.4	21.9
FCOS [29]	ResNet-50	34.9	56.1	37.1
CenterNet [41]	ResNet-101	40.1	63.4	42.4
Ours	ResNet-101	57.6	85.1	65.0

**Table 3 animals-12-01980-t003:** Comparison of fps, AP, and AP50 results of animal segmentation.

Methods	fps	AP	AP50
SGN [41]	1.2	27.3	46.8
Mask R-CNN [21]	2.8	29.6	52.7
PANet [22]	<1	35.8	58.1
Deep Snake [28]	5.3	36.2	59.2

## Data Availability

The data presented in this study are available on request from the corresponding author.

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
