# Peer review of "Contour-Based Wild Animal Instance Segmentation Using a Few-Shot Detector"

_animals, 2022, doi:10.3390/ani12151980_

Round 1

Reviewer 1 Report

I have not worked in this field with respect to terrestrial mammals (but only with sharks) so i am not able to judge literature coverage. This does look like an interesting and useful paper though. There is, however, one major problem which prevented me from making other detailed comments. This is that the English is so poor that i had difficulty in properly understanding and evaluating much of the text.

 Before this paper can be properly evaluated for content, the English needs to be considerably improved and revised by a native English- speaker, so that the scientific points being made are much clearer. Many sentences are simply not expressed the way a native speaker would write then and often the incorrect or inappropriate word is used.  it would take a long time to detail this, but for just one example take the very first sentence: Not only is this basically meaningless (monitoring is a type of study), but in line 8 'supervise' is not the correct word here ( one does not 'supervise' wild animals - maybe you mean 'conserve' or 'manage'?). Line 9  not 'filed' ( that is what one does with paperwork  but 'field'. This goes on on almost every line ( on line 11 for example i think you mean 'recognising' not 'monitoring')

Author Response

Response to respectful Editors and Reviewers

Manuscript Number: animals-1778990

Title: Contour-Based Wild Animal Instance Segmentation by Selecting the optimal Detector

Dear Editors and Reviewer,

Comments:  Before this paper can be properly evaluated for content, the English needs to be considerably improved and revised by a native English-speaker, so that the scientific points being made are much clearer. Many sentences are simply not expressed the way a native speaker would write then and often the incorrect or inappropriate word is used.  it would take a long time to detail this, but for just one example take the very first sentence: Not only is this basically meaningless (monitoring is a type of study), but in line 8 'supervise' is not the correct word here (one does not 'supervise' wild animals - maybe you mean 'conserve' or 'manage'?). Line 9 not 'filed' (that is what one does with paperwork but 'field'. This goes on on almost every line (on line 11 for example I think you mean 'recognizing' not 'monitoring')

Response: Thank you for the comment. The manuscript has been edited by the native speaker. We expect your valuable comments, which will help us to improve the quality of our manuscript.

Reviewer 2 Report

This work is about segmentation of images displaying wild animals trained with small datasets and extendible to animals that do not participate in the training set. This is an ambitious target.

A two-stage model used for segmentation of wild animals is employed. It uses the FSOD CNN as animal species recognition and initial contour detector, and Deep Snake as contour approximation model for instance segmentation of wild mammals.

The dataset used to train their model is constructed from images available in COCO and PASCAL VOC but images are also extracted from documentaries. Using 400/species cannot be considered a very small number of photographs as advertized by the authors although during the results presentation they state that the training photographs were reduced on purpose.

FIg. 2 is barely readable, please increase its resolution because it is very important to understand the development carried out for this work.

Diamond contour in Fig. 3 is also too faint. A more distinct color could be used such as black or yellow and thicker line.

Refining the diamond shape to approach the animal contour needs a much more extensive explanation with plenty of images and examples as well as theory behind it to justify that it is better than eg, thresholding or edge detection followed by edge matching.

How n-Shots work, should also be explained better because there are plenty of results from the use of these methods but their function is unclear.

AP is also undefined although the most important comparison is based on this metric.

The results from the recognition of new species and the estimation of their contour should be restructured to highlight the advantages of this method. Besides some indicative results shown in Fig. 6 and 7 the paper lacks of clear comparison tables that use well defined and appropriate metrics

Author Response

Response to respectful Editors and Reviewers

Manuscript Number: animals-1778990

Title: Contour-Based Wild Animal Instance Segmentation by Selecting the optimal Detector

Dear Editors and Reviewer

Thank you for valuable and constructive comments on our manuscript. We studied these comments carefully and made revisions accordingly.

Comment 1: The dataset used to train their model is constructed from images available in COCO and PASCAL VOC but images are also extracted from documentaries. Using 400/species cannot be considered a very small number of photographs as advertised by the authors although during the results presentation they state that the training photographs were reduced on purpose.

Response: Thank you for the comment. We have revised the interpretation of our dataset and added the website of the public datasets on lines 155-162. We have also supplemented the explanation about the small datasets in lines 88-96, lines 192-199, and 209-215, lines 264-267.

Comment 2: Fig. 2 is barely readable, please increase its resolution because it is very important to understand the development carried out for this work.

Response: Thank you for the comment. We have redrawn Figure 2.

Comment 3: Diamond contour in Fig. 3 is also too faint. A more distinct color could be used such as black or yellow and thicker line.

Response: Thank you for the comment. We have redrawn Figure 3.

Comment 4: Refining the diamond shape to approach the animal contour needs a much more extensive explanation with plenty of images and examples as well as theory behind it to justify that it is better than eg, thresholding or edge detection followed by edge matching.

Response: Thank you for comment. We have added an explanation of the contour approximation method in lines 221-226, 232-233 and we have supplemented the reason that the proposed model is better than thresholding detection, edge detection and data-driven deep learning methods in lines 64-73, and lines 85-96. We have added the experiments as shown in Table 3.

Comment 5: How n-shots work, should also be explained better because there are plenty of results from the use of these methods but their function is unclear.

Response: Thank you for the comment. We have added the explanation of n-shots in Section 2.2.1 and Section 3.1 of the experiment.

Comment 6: AP is also undefined although the most important comparison is based on this metric.

Response: Thank you for the comment. We have provided the meaning of AP in lines 258-262. Because AP is a common metric of object detection methods, and there are many calculation steps, we give the website on AP calculation methods.

Comment 7: The results from the recognition of new species and the estimation of their contour should be restructured to highlight the advantages of this method. Besides some indicative results shown in Fig. 6 and 7 the paper lacks of clear comparison tables that use well defined and appropriate metrics.

Response: Thank you for the comment. In Figure 5, we gave the values of AP, AP50 and AP75 for species detection in small datasets, and we have added the experiments as shown in Table 3 to evaluate the contour approximation method more clearly.

Reviewer 3 Report

The manuscript presents a multiple  neural network-based system for wild animal instance segmentation in still images, aimed to the recognition of the animal's countour, so that it could be used later for behavior analyses. The system is composed of two stages, the first one aimed to the detection of the animal in the image which makes use of a Few-Shot Object Detection framework, and the second aimed to finding the animal's contour which relies on Deep Snake techniques.

This document is mostly well written, with few English language issues, easy to read and to understand. It is organized as follows: Abstract, a first Section containing a short Introduction, a second Section on the Materials and Methods followed, a third Section on the Results, a fourth Section including the Conclusions of the work, and finally the References used.

As it is, In my opinion the manuscript is of some value, offering an interesting contribution to the identification an definition of animals' contours on still images.

Aiming to improve the quality of the work and without any intent to underrate neither its accuracy nor its contributions, I would like to make the following suggestions:

1. Please, check the following potential issues that I consider necessary to:

LINE                  READS                                                 SUGGESTION
20                      'study'                                                'studying'
21                      'dataset'                                             'datasets'
46                      'only study'                                         'only make possible to study'
50                      'First'                                                   'first'
51                      'whch was obviously'                          'which obviously'
54                      'exists many deep learning based'      'exist many deep learning-based'
54                      'and meanwhile'                                  Please, reconsider rewriting this clause to make it more clear
56                      'Defect'                                                'defect'
66                      'Inception'                                            'inception'
68                      'detection of the images'                     'detection in the images'
69                      'in instance'                                          'to the instance'
78                      'efforts have'                                        'effort has'
79-80                 'On the other hand ...'                          Please, reconsider rewriting this clause to make it more clear
88                      'of small'                                              'of the small'
92                      'threeefold.'                                         'threefold as explained next.'
99                      'new the species'                                   'new species'
100                     'dataset and small-size dataset'           'datasets and small-size datasets'
110                     'give'                                                   'provide'
115                      'include the limited'                           'include a limited'
123                      '4884'                                                 '4,884'
222-223               'The comparison ...'                              Please, reconsider rewriting this sentence to make it more clear
250                      'images,'                                              'images;'
258                      '3.1.3. ...'                                               Please, correct the indentation

2. I would recommend rewriting the title to better describe the goals of the work, as the 'by selecting the optimal detector' adds some confusion to it.

3. Please, make clear what do you mean with: 'and even identify the new species not included in the training set' (line 89)

4. In Sub-section 2.1., what are the characteristics (resolution, color depth, encoding, etc.) of the images in the different datasets?

5. Please clarify the intended use and characteristics of the 'Mammate' and the 'MammData' datasets mentioned in line 122.

6. Regarding the above item, which one of those two is the dataset mentioned in the labels of Figure 1 and Table 1?

7. Please use the corresponding points in the abbreviations in the headers of Table 1.

8. Please state the different data flows within the proposed structure. Also, highlight the changes in the animal image shown, as they seem very much similar in all four insertions.

9. The description provided in section 2 is too simplistic and it does not provide enough information to analyze in depth its operation, appropriateness and expected results. This description should be extended enough to help to understand the solution being proposed in the manuscript.

10. The first part of Section 3 seems to fit better in the Materials and Methods Section, or else on a new Case Study Section.

11. The paragraphs in lines 201-205 requieres a more detailed explanation with appropriate justification of the decissions made on the different parameters mentioned.

12. As mentioned in point #3 above, the 'Detecting new species' in lines 213 and following requires of a more extense explanation regarding its interpretation and intended operation. See also lines 234-237 on this issue.

13. Figures 4 and 5: Please check the format of the labels.

14. Please clarify the meaning of the sentence in lines 245-247.

15. All in all, Section 3 seems not to provide a quantitative benchmark of the proposed methodology versus the alternative ones available, nor includes a clear explanation on its caveats and limitations. Additionally, the writing is too dense and it is difficult to follow the text; please, consider rewriting it for a better understanding.

16. In my opinion, the Conclusions section is somehow short, and would require to rewrite it for accuracy (e.g., it is not mentioned that the 'instance segmentation of wildlife' is performed on still images).

Author Response

Response to respectful Editors and Reviewers

Manuscript Number: animals-1778990

Title: Contour-Based Wild Animal Instance Segmentation by Selecting the optimal Detector

Dear Editors and Reviewer,

Thank you for valuable and constructive comments on our manuscript. We studied these comments carefully and made revisions accordingly.

Comment 1: Please, check the following potential issues that I consider necessary to.

Response: Thank you for the comment. I'm sorry for so many spelling mistakes. We have carefully revised the above English expressions. The manuscript has been edited by the native speaker.  

Comment 2: I would recommend rewriting the title to better describe the goals of the work, as the 'by selecting the optimal detector' adds some confusion to it.

Response: Thank you for the comment. We have revised the title of the paper.

Comment 3: Please, make clear what do you mean with: 'and even identify the new species not included in the training set' (line 89).

Response: Thank you for the comment. We have revised this expression in the manuscript in lines 100-103. We train the ability of feature extraction with the base classes that contains abundant images. After that, we utilize the learning ability of the model to recognize the new species that are not included in the base classes, and the new species only have a small number of training samples.

Comment 4: In Sub-section 2.1., what are the characteristics (resolution, color depth, encoding, etc.) of the images in the different datasets?

Response: Thank you for the comment. We have added the characteristics of images in the dataset, such as resolution, color depth, etc., in lines 159-162.

Comment 5: Please clarify the intended use and characteristics of the 'Mammate' and the 'MammData' datasets mentioned in line 122.

Response: Thank you for the comment. We have modified the spelling mistake of the word 'Mammate', so we have replaced the word 'Mammate' with ‘MammData’. The dataset of wild mammal constructed in this manuscript is named MammData, which is used for model training and testing. A detailed explanation of this dataset is given in lines 155-162, Figure 1 and Table 1.

Comment 6: Regarding the above item, which one of those two is the dataset mentioned in the labels of Figure 1 and Table 1?

Response: Thank you for the comment. Sorry, 'Mammate' is misspelled, we have changed it to 'MammData'.

Comment 7: Please use the corresponding points in the abbreviations in the headers of Table 1.

Response: Thank you for the comment. We have modified the header of Table 1.

Comment 8: Please state the different data flows within the proposed structure. Also, highlight the changes in the animal image shown, as they seem very much similar in all four insertions.

Response: Thank you for the comment. The four pictures respectively express the gradual approximation process of the contour. In order to express it more clearly, we have thickened the contour.

Comment 9: The description provided in section 2 is too simplistic and it does not provide enough information to analyze in depth its operation, appropriateness and expected results. This description should be extended enough to help to understand the solution being proposed in the manuscript.

Response: Thank you for the comment. We have supplemented the explanation of FSOD in lines 88-96, lines 192-199, and 209-215. and Deep Snake in lines 221-226, 232-233. The references about these two methods are also given.

Comment 10: The first part of Section 3 seems to fit better in the Materials and Methods Section, or else on a new Case Study Section.

Response: Thank you for the comment. We have transferred this part to Section 2.1 'Materials'.

Comment 11: The paragraphs in lines 201-205 requires a more detailed explanation with appropriate justification of the decisions made on the different parameters mentioned.

Response: Thank you for the comment. We have supplemented and explained the setting method of parameters in Section2.1 'Materials'.

Comment 12: As mentioned in point #3 above, the 'Detecting new species' in lines 213 and following requires of a more extense explanation regarding its interpretation and intended operation. See also lines 234-237 on this issue.

Response: Thank you for the comment. We have explained the meaning of new species in detail and have also supplemented the explanation about the small datasets in lines 88-96, lines 192-199, 209-215, and lines 264-267.

Comment 13: Figures 4 and 5: Please check the format of the labels.

Response: Thank you for the comment. We have redrawn Figures 4 and 5.

Comment 14: Please clarify the meaning of the sentence in lines 245-247.

Response: Thank you for the comment. We have reinterpreted these sentences in lines 274-275.

Comment 15: All in all, Section 3 seems not to provide a quantitative benchmark of the proposed methodology versus the alternative ones available, nor includes a clear explanation on its caveats and limitations. Additionally, the writing is too dense and it is difficult to follow the text; please, consider rewriting it for a better understanding.

Response: Thank you for the comment. We have provided the meaning of AP on lines 258-262. Because AP is a common metric of object detection methods, and there are many calculation steps, we give the website on AP calculation methods. In Figure 5, we gave the values of AP, AP50 and AP75 for species detection in small datasets. We have also added the experiments as shown in Table 3 to evaluate the contour approximation method more clearly. We have redrawn the Figure 8 and rewritten the explanation of Figure 8 for a better understanding, as seen in lines 330-346, and lines 348-351.

Comment 16: In my opinion, the Conclusions section is somehow short, and would require to rewrite it for accuracy (e.g., it is not mentioned that the 'instance segmentation of wildlife' is performed on still images).

Response: Thank you for the comment. We have added the limitations of the paper research and future work in lines 363-371.

Round 2

Reviewer 2 Report

The authors provided an improved revised paper where many of my comments were addressed. For example a clearer explanation was given about AP, n-Shot, etc. However there are still some issues that have not been addressed:

- In AP50, AP75 I still cannot understand "threshold matching" at least in the single sentence where this is explained in lines 261-262

- FSOD is the basic pillar of the article but I am a bit confused about its difference with Transfer Learning which similarly uses models trained for some categories and they are retrained for new categories with few new samples. It would be interesting to highlight the difference

- More information is also needed about how FSOD is realized (which parts of the architecture of Fig. 2 are used to implement FSOD and how this is achieved

Reviewer 3 Report

The Authors made an effort to modify the manuscript to address the issues mentioned in my initial review, with success. They have also provided a reply letter to answer to the indications provided to them after the review.

After reviewing the new version of the manuscript, I would like to make the following comments and recommendations to the Authors, aiming to improve the quality of the document so it meets the minimum requirements for that:

1. Please check the uniformity of the font in the title of the work.

2. In line 106, consider removing the bold type.

3. In line 158, consider removing the space before the ')'. Same for line 260.

4. In the bottom line of Table 1, please use ',' for separating the thousand digit blocks.

5. In line 276, consider replacing 'As seen in Figure 5,' with 'As seen in that figure'.

6. In line 301, consider replacing 'because that the' with 'replacing the'.

7. In line 370, consider replacing 'videos, we' with ''videos. We'.

Round 3

Reviewer 2 Report

The authors added new figures and explanations about the differences of FSOD and Transfer learning as I asked

Reviewer 3 Report

In my opinion, the document with the changes made cam be recommended for publication.